# An Equity and Justice-Informed Ethical Framework to Guide Incidental Findings in Brain Imaging Research

Sonu M. M. Bhaskar [1,2,3,4,5,*]

1    Global Health Neurology Lab, Sydney, NSW 2000, Australia; sonu.bhaskar@globalhealthneurolab.org;
     Tel.: +61-(02)-873-89179; Fax: +61-(02)-8738-3648
2    Neurovascular Imaging Laboratory, Clinical Sciences Stream, Ingham Institute for Applied Medical Research,
     Liverpool, NSW 2170, Australia
3    Department of Neurology & Neurophysiology, Liverpool Hospital & South West Sydney Local Health
     District (SWSLHD), Liverpool, NSW 2170, Australia
4    NSW Brain Clot Bank, NSW Health Pathology, Sydney, NSW 2170, Australia
5    Stroke & Neurology Research Group, Ingham Institute for Applied Medical Research,
     Liverpool, NSW 2170, Australia

**Abstract:** The handling of incidental findings (IFs) in brain imaging studies has been a source of contention among scientists and bioethicists. A conceptual framework informed by diversity, equity, and inclusion (DEI) and distributive justice approaches, namely EUSTICE, is proposed for the ethical handling and reporting of IFs in brain imaging research. I argue that EUSTICE provides a systematic and inclusive approach to addressing the ethical conundrum around IF disclosure and managing IFs proportionately and sensitively in brain imaging research. The EUSTICE framework may have implications for the field of neurosciences or human studies broadly in guiding ethics of IFs in research.

**Keywords:** ethics; brain imaging; neuroscience; radiology; framework

## 1. Introduction

Brain imaging has gained increasing popularity among neuroscientists and clinicians, including academic radiologists, as a tool to study brain structure and function, as well as to aid diagnosis and prognosis and guide the treatment of brain diseases [1–3]. Even during COVID-19, brain imaging techniques such as magnetic resonance imaging (MRI) were used to detect brain abnormalities [4]. Researchers and bioethicists have been debating how to handle incidental findings (IFs) in brain imaging research studies [5,6]. There is a growing consensus that there should be a pathway to appropriately address the discovery of IFs in research settings [6–8]. Previously, efforts have been made to provide practical recommendations or pathways to manage IFs in human research studies in general and brain imaging studies in particular [9–11]. Notably, the approach to manage IFs, as proposed by Illes et al., offers a practical solution that allows but does not mandate the assessment of neuroimaging for clinically significant observation by a medical professional [11]. Though, in the research setting, given the limited reporting and ambiguity around the analysis and reporting of IFs, this limits the IF review at the disposal of individuals and may undermine the interests of patients, especially those from vulnerable backgrounds, as well as the transparency of peer review of IF and its clinical implications with representation from various stakeholders involved in a research study. It can also be argued that the said process may be construed as paternalistic, lacking a consultatory decision-making approach, which historically often puts vulnerable populations at risk or disadvantage [12], and lacking sensitivity to principles of distributive justice. The current article presents an Equity- and JUSTICE-informed, namely "EUSTICE", conceptual framework for ethical handling and reporting of IFs in brain imaging research, and its various elements are discussed.

## 2. Incidental Findings in Brain Imaging Research

Incidental findings refer to any information that is discovered during the course of a medical or research study that is not directly related to the primary purpose of the study [13]. In the context of brain imaging, incidental findings are abnormalities that are discovered on a brain scan that were not the primary focus of the scan. These findings can be discovered during a variety of brain imaging studies, including magnetic resonance imaging (MRI), computed tomography (CT) scans, and positron emission tomography (PET) scans. Incidental findings in brain imaging studies are relatively common, and they can range from minor abnormalities that are of little clinical significance to more serious conditions that require further evaluation and treatment. According to a meta-analysis of IF prevalence in brain MRI research, IFs are frequent, with prevalence estimates ranging from 5–20% [14]. Some examples of incidental findings that may be discovered in brain imaging studies include: (a) lesions or abnormalities in the brain tissue, such as tumors, cysts, or scars; (b) abnormalities in the size or shape of the brain or its structures; (c) abnormalities in the blood vessels or circulation in the brain; (d) abnormalities in the flow of cerebrospinal fluid, and (e) abnormalities in the metabolism or function of brain cells. IFs in brain imaging research may have broader ethical implications for the field of neuroscience broadly. For example, the discovery of an abnormal brain structure or function in a participant may have implications for our understanding of brain development or disease processes [14].

The ethical implications of incidental findings in brain imaging research are complex and multifaceted, and they raise important questions about the responsibilities of researchers, the rights of research participants, and the appropriate handling of sensitive medical information. One of the key ethical issues surrounding incidental findings in brain imaging research is the responsibility of researchers to disclose this information to participants. In many cases, brain imaging studies are conducted for research purposes and do not involve a medical diagnosis or treatment plan. This means that participants may not be expecting to receive any information about their health status or potential medical conditions. This information may be sensitive and potentially embarrassing, and researchers have a responsibility to protect it and ensure that it is not misused or disclosed without the participant's consent. However, if a researcher discovers an incidental finding during the course of a study, they may have an ethical obligation to share this information with the participant. Special ethical considerations may apply to subjects participating in brain imaging research, including [14–17], but not limited to young adults [18], pediatric subjects [19], and other vulnerable populations such as ethnic minorities and indigenous people [20].

Awareness of clinically relevant IFs may guide research participants to seek appropriate treatment or take preventative action [21]. Unfortunately, disclosure of IFs may have potentially devastating consequences, especially when the clinical relevance of IFs is low or debatable [22]. False-positive diagnoses are not uncommon and are linked to potential harms or negative impacts, including psychological distress, overdiagnosis, health risks, and costs [23–25].

## 3. Ethical Aspects concerning Incidental Findings in Neuroscience Research

Researchers may have a responsibility to disclose any IFs to participants, to handle sensitive medical information with care, and to consider the potential broader impact of their research on the scientific community. By taking these considerations into account, researchers can ensure that their work is conducted ethically and responsibly. Overall, handling IFs in a fair and just manner involves treating all patients with respect and dignity and providing them with the necessary information and resources to make informed decisions about their healthcare. There are several key considerations surrounding principles of human research ethics that can guide researchers in handling IFs [26]: (a) Respect for autonomy, informed consent, and confidentiality: Researchers have a duty to respect the autonomy of research participants and to consider their interests and preferences when making decisions about how to handle IFs. Patients should be fully informed about the potential risks, benefits, and alternatives to any medical procedure or examination, includ-

ing the possibility of IFs. This helps ensure that patients can make informed decisions about their care. The confidentiality of incidental findings should be respected, and the information should only be shared with those who need to know. (b) Nonmaleficence and disclosure: Researchers have a duty to do no harm and should consider the potential risks and benefits of disclosing IFs to participants. This includes not withholding information that may be of benefit to research participants. It is important to disclose incidental findings to the patient in a timely and appropriate manner. This may involve discussing the findings with the patient, providing them with written information, providing appropriate follow-up care, or referring them to a specialist for further evaluation. (c) Beneficence: Researchers have a duty to act in the best interests of research participants and should consider whether disclosing IFs would be beneficial to the participant. (d) Justice and quality of care: Researchers should ensure that the handling of IFs is fair and equitable and that any decisions made are not biased or discriminatory. This includes ensuring that the benefits and burdens of research are distributed fairly among all participants. Patients should receive high-quality care regardless of their ability to pay or other social factors. Consider the potential financial impact of follow-up care and treatment on the patient, and work with the patient to identify options for financing this care.

Before introducing the EUSTICE framework, the following sections will provide a background on distributive justice and DEI concepts and their application to human/medical research and ethics, which is at the foundation of the proposed framework.

*Distributive justice:* Distributive justice is a concept in political philosophy that refers to the fair and equitable distribution of resources, benefits, and burdens within a society [27]. It is a fundamental aspect of any just and fair society, as it determines how the benefits and burdens of social, economic, and political systems are distributed among the members of a community. There are various theories of distributive justice that attempt to address the question of how resources should be distributed within society. One of the most well-known theories is Rawls' theory of justice [28], which states that the distribution of resources should be such that it would be accepted by rational actors in an original position of equality [29]. Rawls argues that, in order to determine a fair distribution of resources, we must imagine ourselves in a hypothetical "original position" where we are free and equal but do not know our own place in society, our natural abilities, or our social status. From this position, we would choose the principles of justice that would govern the distribution of resources in our society, including health and social determinants of health [30]. According to Rawls, the principles of justice that we would choose would be: (a) each person is to have an equal right to the most extensive total system of equal basic liberties compatible with a similar system of liberty for all, and (b) social and economic inequalities are to be arranged so that they are both: (a) reasonably expected to be to everyone's advantage, and (b) attached to positions and offices open to all. Another theory of distributive justice is the libertarian theory [31], which states that individuals have a natural right to own and control their own property and that the state has a limited role in redistributing wealth. According to this theory, the state should not interfere with individuals' rights to own and control their own property, and any redistribution of wealth should be voluntary rather than mandatory. There are also various other theories of distributive justice, such as the socialist theory, which advocates for the collective ownership of the means of production and the distribution of resources according to the needs of the community; the Rawlsian difference principle, which argues that social and economic inequalities should be allowed as long as they benefit the least advantaged members of society; and the capabilities approach, which focuses on the abilities and opportunities that individuals have to lead a fulfilling life, rather than simply the distribution of resources.

The concept of distributive justice is closely related to the field of human research ethics, as it deals with the fair distribution of the benefits and burdens of research among different groups of people [32]. To ensure that human research is conducted ethically, it is important to consider the principles of distributive justice and ensure that the distribution of the benefits and burdens is fair and just. One way in which the principles of distributive

justice can be applied to human research ethics is by ensuring that the benefits of the research are distributed fairly among different groups of people [33]. For example, if a particular research study is being conducted on a new medical treatment, it is important to ensure that the treatment is made available to all those who need it, regardless of their social or economic status. This means that the treatment should not be restricted to certain groups of people, such as those who can afford to pay for it, while others are left out.

Another way in which distributive justice can be applied to human research ethics is by ensuring that the burdens of research are distributed fairly among different groups of people. This includes ensuring that the risks of participating in a research study are minimized and that participants are fully informed about the potential risks and benefits of participating in the study. It is also important to ensure that the selection of research subjects is fair and that subjects are not unfairly disadvantaged or excluded based on their social or economic status. In addition to these considerations, it is important to ensure that the distribution of benefits and burdens from research is transparent and open to public scrutiny. This helps to ensure that the distribution of benefits and burdens is fair and just and that any potential inequities are identified and addressed. Overall, the concept of distributive justice is an important consideration in the field of human research ethics, as it helps to ensure that the benefits and burdens of research are distributed fairly among different groups of people. By applying the principles of distributive justice to human research, we can help to ensure that research is conducted in an ethical and just manner and that the benefits of research are shared fairly among all members of society.

*Diversity, Equity, and Inclusion:* Diversity, equity, and inclusion (DEI) theory is a framework that aims to promote fairness and justice in organizations and communities by promoting the inclusion and representation of diverse groups of people [34]. Diversity refers to the range of differences among individuals, including differences in race, ethnicity, gender, sexual orientation, age, ability, religion, and cultural background. Equity refers to the fair treatment of individuals and groups, including the fair distribution of resources, opportunities, and privileges. Inclusion refers to the active engagement and welcoming of diversity within an organization or community.

DEI theory recognizes that diversity and inclusion are essential to creating a just and equitable society, as they help to promote understanding, respect, and appreciation for differences among individuals and groups. DEI theory also acknowledges that historically marginalized and underrepresented groups often face barriers and discrimination in accessing resources, opportunities, and privileges, and therefore efforts to promote equity are necessary to address these inequalities [35]. It also involves acknowledging and celebrating the unique perspectives, experiences, and contributions of all individuals.

To promote DEI, organizations and communities can adopt policies and practices that actively seek to increase diversity, promote equity, and create inclusive environments. This may include things like implementing hiring and promotion practices that promote diversity, providing resources and support for marginalized groups, and creating safe and welcoming spaces for all members of the community. By adopting DEI principles and practices, organizations and communities can work towards creating a more just and equitable society for all.

The principles of DEI can be applied to the field of human research ethics to ensure that research is conducted in a manner that is fair and just and that the benefits of research are shared equitably among all members of society [36]. One way in which DEI principles can be applied to human research ethics is by ensuring that the selection of research subjects is diverse and representative of the population being studied [37]. This includes considering factors such as race, ethnicity, gender, sexual orientation, age, ability, and socio-economic status to ensure that the research sample is representative of the larger population. It is also important to ensure that underrepresented groups are not unfairly excluded from participating in research.

In addition to ensuring diversity in the selection of research subjects, it is also important to ensure that the benefits of research are shared equitably among all members

of society. This may involve making the results of research available to all, regardless of their social or economic status, or ensuring that the benefits of research are distributed in a manner that is fair and just. DEI principles can also be applied to the design and conduct of research studies in order to ensure that the research process is inclusive and respectful of the needs and perspectives of all research subjects [37]. This may involve consulting with and involving underrepresented groups in the design and conduct of research and ensuring that the research process is transparent and open to public scrutiny. Overall, the principles of DEI can be applied to the field of human research ethics in order to ensure that research is conducted in a fair and just manner and that the benefits of research are shared equitably among all members of society. By applying DEI principles to human research, we can help to create a more diverse, equitable, and inclusive society [38].

### 4. Towards a Conceptual Ethical Framework for Handling Incidental Findings in Brain Imaging Research

The two opposing arguments around disclosure or no disclosure of IFs raise important issues for ethical debate. The controversy often distills to two lingering issues: (a) what is the responsibility of researchers when the clinical benefit of IFs disclosure is low, uncertain, or unknown, which is often the case? and (b) what about the risks of unnecessary harm to patients especially those who are young or come from vulnerable backgrounds? To address these concerns, I propose an integrated EUSTICE framework (Figure 1), underpinning diversity, equity, and inclusion (DEI) [39] and distributive-justice-[40–42] informed approaches towards handling and reporting IFs in neuroscience or brain imaging research so that IFs are managed proportionally and sensitively in a way that is not resource-demanding or cost-inhibitive to pursue and conduct brain imaging research [43]. Notably, the EUSTICE framework is rooted in the Rawlsian theory of distributive justice. I draw upon the distributed justice concept that institutions and researchers have a responsibility to perform research in a way consistent with the state's obligation of distributive justice to provide access to basic healthcare to all its citizens and its duty to support citizens belonging to vulnerable backgrounds and low-resourced settings [40,42]. I postulate that fostering a diverse, equitable, and inclusive structure and engagement is key to realizing a distributive justice-informed ethical framework [39,44]. Individuals belonging to certain communities have historically faced barriers to access and participation in research or clinical trials [45–48]. Besides, significant disparities exist in access and availability of healthcare, as well as health outcomes, among vulnerable populations, including ethnic minorities, human immunodeficiency virus (HIV) patients [49] and indigenous populations, among others [46,50–52]. I argue that by integrating DEI in all aspects of the ethical framework, the implementation of distributive justice can be done inclusively and is likely to be trusted by research participants of all backgrounds.

Various elements of the proposed EUSTICE ethical framework informed by DEI and Justice approaches are discussed below.

(1) Researchers are obligated to look for and disclose IFs only within the remit of basic care that research participants are entitled to per distributive justice principles [42]. It is not required for researchers to screen for brain abnormalities on brain imaging nor are they obligated to follow up as this is not considered basic care as per the ethical principle of distributive justice [42].

(2) Research studies should have a distinct IFs committee comprising 1–2 investigators involved in the research study (including the chief/principal investigator), bioethicists, community representatives, including those belonging to marginalized backgrounds, as well as a neurologist (or a general physician) and clinical neuroradiologist not directly involved in the research study.

(3) All cases, or matters thereof, of IFs should be referred to the above committee. The matter of disclosing IFs should be considered considering the following matters: (a) clinical significance of IFs (this should factor in the severity of IFs and health risks); IFs with low, uncertain, or unknown clinical significance are not disclosed. Clinically

significant IFs should be disclosed to the participant through their respective health-care provider or family physician (FP), unless the research subject has refused such IFs [53], who can assess the disclosure considering the clinical history of the patient, which is privy to the severity of IFs and associated health risks. The adjudication or discussion on the clinical significance of IFs should be based on the standards of care or current recommendations or indications at the time of review of the merit of the case by the IF committee. (b) DEI considerations: Should there be no FP or healthcare provider, which may be the case for members of vulnerable communities, the disclosure of clinically significant IFs should be made directly to the concerned research subject [53]. Besides, the committee may also consider DEI issues as they may apply on a case-to-case basis. For example, a person without an FP can be linked to an FP in close vicinity so that the IFs disclosure can then be made through the FP.

(4)     Above information or policy about handling IFs and disclosure should be incorporated into the informed consent forms and research information booklet. Research participants should have the known risks of IFs explained. Participants should also be asked if they want to be informed about IFs. The consent form may also include a clause indicating exemption for legal liabilities to consequences of incidental findings [13].

(5)     Data on the prevalence and severity of IFs should be included in research publications. Furthermore, the institutional research ethics board should be informed about IFs.

(6)     Standardized education toolkits for discovery, reporting, ethical aspects, and disclosure surrounding IFs in brain imaging research should be developed.

(7)     All researchers involved in the study, and members of the IF committee, should be provided education and training on IFs and communication skills on various considerations that may apply to the disclosure of IFs [13].

## EUSTICE Framework for Incidental Findings in Brain Imaging Research

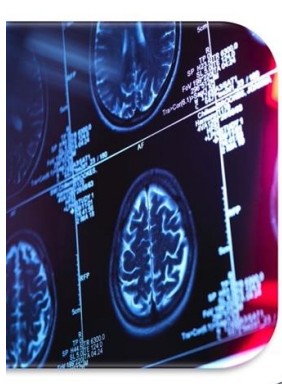

**Step 1: Information on handling incidental findings (IFs) and disclosure is provided in the informed consent forms. Known risks of IFs to be explained to subjects.**\*

*\*Research subjects to be asked if they want to be informed on IFs. The consent form may also include a clause indicating exemption for legal liabilities to consequences of IFs.*

**Step 2: IFs discovered by the researcher. Sent to the IFs committee for review.**

**Step 3: Decision on IFs disclosure to research subject based on clear clinical significance of IFs.**

**Step 4: IFs with low, uncertain, or unknown clinical significance are not disclosed.**

**Step 5: Clinically significant IFs disclosed to research subject through their respective healthcare provider or family physician.**\*\*

*\*\*In case of no family physician, the disclosure on clinically significant IFs made directly to the research subject. The IFs Committee may also consider Diversity, Equity and Inclusion (DEI) and distributive justice issues as they may apply on a case-to-case basis.*

**Figure 1.** Proposed ethical framework on handling and disclosure of incidental findings in brain imaging research informed by diversity, equity, and justice, and distributive justice principles.

Researchers should have a plan in place for handling incidental findings before the study begins and should be transparent about their approach with participants. This should include obtaining explicit consent from participants to disclose any incidental findings and providing resources and support to participants who may be affected by the disclosure

of this information. Overall, it is important to handle incidental findings in a way that is transparent, ethical, and fair, taking into account the rights and needs of the patient.

## 5. Conclusions

Researchers can address ethical concerns surrounding IFs by developing clear policies and procedures for handling incidental findings and sensitive medical information. Researchers have a responsibility to consider the potential impact of these findings on the larger scientific community and to carefully consider the ethical implications of their research. The proposed ethical framework, EUSTICE, informed by DEI and distributive justice principles, may provide a systematic, inclusive, and just approach to handling and disclosure of IFs in brain imaging research. It is postulated that EUSTICE will provide a research participant-centered ethical framework to manage IFs in brain imaging research proportionately and sensitively so that the process is not resource-intensive and inhibitive cost-wise.

**Funding:** This research received no funding.

**Institutional Review Board Statement:** Not applicable.

**Informed Consent Statement:** Not applicable.

**Data Availability Statement:** The original contributions presented in the study are included in the article, and further inquiries can be directed to the corresponding author.

**Acknowledgments:** The content is solely the responsibility of the authors and does not necessarily represent the official views of the affiliated/funding organization/s.

**Conflicts of Interest:** The authors declare that they have no conflict of interest.

## Abbreviations

| | |
|---|---|
| MRI | magnetic resonance imaging |
| EUSTICE | equity and justice |
| IFs | incidental findings |
| HIV | human immunodeficiency virus |
| DEI | diversity, equity, and inclusion |
| FP | family physician |

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
