# Peer review of "An Equity and Justice-Informed Ethical Framework to Guide Incidental Findings in Brain Imaging Research"

_clinpract, doi:10.3390/clinpract13010011_

Round 1

Reviewer 1 Report

Dear authors

The purpose of the study to provide a participant-centered ethical research framework for managing IFs in neuroscience research, thus, is relevant and should be addressed in more depth at each point cited by the authors.

The manuscript needs more robust adjustments in the discussion of each point raised by the authors, being a very synthetic discussion

Best regards!

Author Response

Reviewer 1

Dear authors

The purpose of the study to provide a participant-centered ethical research framework for managing IFs in neuroscience research, thus, is relevant and should be addressed in more depth at each point cited by the authors.

The manuscript needs more robust adjustments in the discussion of each point raised by the authors, being a very synthetic discussion

Best regards!

Reply# We thank the reviewer for consideration of our work and valuable. We have expanded the discussion to broadly give a background of various aspects of the framework especially the distributive justice and DEI principles. The updated manuscript with track changes have been provided. Besides, Figure 1 has also been updated.

Reviewer 2 Report

This is an interesting paper on a timely topic.

That being said, it is more than a little half-baked.

First and foremost, the author makes no effort to address the issue of snowballing bureaucratic impediments to the pursuit of science. Research Ethics Boards (REBs) or Institutional Review Boards (IRBs), depending on jurisdiction, already review studies. Why don't we simply task them with considering incidental findings (IFs) instead of establishing a new IFs committee?

Now, an argument could be made for this, but the author provides none.

Second, I have some concerns about the lack of engagement with prior research on IFs in neuroimaging. How is this proposal better than proposals for managing IFs (notably "opt-out" or "opt-in" to disclosure) put forward by Illes?

Some references to engage with on this front include:

https://scholar.google.com/citations?view_op=view_citation&hl=en&user=XAhJjMUAAAAJ&citation_for_view=XAhJjMUAAAAJ:Y0pCki6q_DkC

https://scholar.google.com/citations?view_op=view_citation&hl=en&user=XAhJjMUAAAAJ&cstart=20&pagesize=80&citation_for_view=XAhJjMUAAAAJ:ULOm3_A8WrAC

https://scholar.google.com/citations?view_op=view_citation&hl=en&user=XAhJjMUAAAAJ&cstart=20&pagesize=80&citation_for_view=XAhJjMUAAAAJ:7Hz3ACDFbsoC

Also, it is not clear how steps 5 and 6 (in Figure 1) differ. A lot of the text is repeated word for word. In fact, the whole figure is not really clear. Does the author want to say that IFs Committee makes case-by-case decisions based on their take on the patient's race, gender and ethnicity and completely ignoring patient preferences? Again, an argument could be made for establishing the consultatory (and not paternalistic) role of IFs committee, but the reader is left wondering at the possibility of abuse should such a system be established, while the author offers no clarity in this regard.

Last but not least, the author hints at substantive content in the domain of social justice, while failing to engage with any primary literature on the issue. Sure, there are references to some recent work applying social justice in the healthcare context (Graham et al 2021 is cited three times with three separate refs: 23, 34 and 35), but it is unclear if the author favors the utilitarian version of social justice (as Graham and others do) or if the substantive content of justice is captured by Rawlsian principles or enumerated by overarching requirements of equality, merit and need (following David Miller). In fact, in order to have a conceptual framework in more than name only, the author needs to specify what it entails - otherwise the proposal boils down to "handwaving" at best and to "fairwashing" for blatant paternalism at worst.

Author Response

We thank the reviewer for the comments. We provide a point by point rebuttal to each of the comments.

C#1: This is an interesting paper on a timely topic.

Reply# We thank the reviewer for the review of the manuscript and several valuable suggestions which have helped us to significantly improve the manuscript and provide improved clarity on various aspects of this work.

C#2: That being said, it is more than a little half-baked.

First and foremost, the author makes no effort to address the issue of snowballing bureaucratic impediments to the pursuit of science. Research Ethics Boards (REBs) or Institutional Review Boards (IRBs), depending on jurisdiction, already review studies. Why don't we simply task them with considering incidental findings (IFs) instead of establishing a new IFs committee.

Now, an argument could be made for this, but the author provides none.

Reply# We thank the reviewer for the suggestion. Unfortunately, given the hundreds or thousands of ethics applications and projects which are overseen by the REB/IRB, as well as the volume of such referrals, bureaucratic challenges and time constraints, I believe it will be rather difficult to task IRBs to establish such a committee. This should be done at a project level and off course reports on IFs can also be submitted as part of the regular updates to the IRB during the course of the study. We also fear given the bureaucracy and time limitations, such a committee within the IRBs is likely to delay the studies or projects. We have made significant changes to expand the theoretical basis of this work.  

C#3: Second, I have some concerns about the lack of engagement with prior research on IFs in neuroimaging. How is this proposal better than proposals for managing IFs (notably "opt-out" or "opt-in" to disclosure) put forward by Illes?

Some references to engage with on this front include:

https://scholar.google.com/citations?view_op=view_citation&hl=en&user=XAhJjMUAAAAJ&citation_for_view=XAhJjMUAAAAJ:Y0pCki6q_DkC

https://scholar.google.com/citations?view_op=view_citation&hl=en&user=XAhJjMUAAAAJ&cstart=20&pagesize=80&citation_for_view=XAhJjMUAAAAJ:ULOm3_A8WrAC

https://scholar.google.com/citations?view_op=view_citation&hl=en&user=XAhJjMUAAAAJ&cstart=20&pagesize=80&citation_for_view=XAhJjMUAAAAJ:7Hz3ACDFbsoC

Reply# We thank the reviewer for suggesting these studies for discussion. We have included the following statement to clarify this in the introduction.

Page 1-2: Lines 35-50

Previously, efforts have been made to provide practical recommendations or pathways to manage IFs in human research studies in general and brain imaging studies in particular [9-11]. Notably, the approach to manage IFs as proposed by Illes et al offers a practical solution that allows but doesn’t mandate, assessment of neuroimaging for clinically significant observation by a medical professional [11]. Though, in the research setting, given the limited reporting and ambiguity around the analysis and reporting of IFs, this limits the IF review at the disposal of individual and may undermine the interests of patients, especially from vulnerable backgrounds, as well as transparency of peer-review of IF and its clinical implication with representation from various stakeholders involved in the research study. Besides, it can also be argued that the said process may be construed as paternalistic, lacking a consultatory decision-making approach, which historically often puts vulnerable populations at risk or disadvantage [12], and lacking sensitivity to principles of distributive justice. The current article presents an Equity and jUSTICE informed, namely “EUSTICE”, conceptual framework for ethical handling and reporting of IFs in brain imaging research has been proposed and its various elements are discussed.

C#4: Also, it is not clear how steps 5 and 6 (in Figure 1) differ. A lot of the text is repeated word for word. In fact, the whole figure is not really clear. Does the author want to say that IFs Committee makes case-by-case decisions based on their take on the patient's race, gender and ethnicity and completely ignoring patient preferences? Again, an argument could be made for establishing the consultatory (and not paternalistic) role of IFs committee, but the reader is left wondering at the possibility of abuse should such a system be established, while the author offers no clarity in this regard.

Reply# We thank the reviewer for highlighting the redundant sentence in the Figure 1. The Figure has now been updated and the redundant parts have already been removed. Besides, an argument to support formation of such a committee has been provided as above.

Besides, it can also be argued that the said process may be construed as paternalistic, lacking a consultatory decision-making approach, which historically often puts vulnerable populations at risk or disadvantage [12], and lacking sensitivity to principles of distributive justice.”

C#5: Last but not least, the author hints at substantive content in the domain of social justice, while failing to engage with any primary literature on the issue. Sure, there are references to some recent work applying social justice in the healthcare context (Graham et al 2021 is cited three times with three separate refs: 23, 34 and 35), but it is unclear if the author favors the utilitarian version of social justice (as Graham and others do) or if the substantive content of justice is captured by Rawlsian principles or enumerated by overarching requirements of equality, merit and need (following David Miller). In fact, in order to have a conceptual framework in more than name only, the author needs to specify what it entails - otherwise the proposal boils down to "handwaving" at best and to "fairwashing" for blatant paternalism at worst.

Reply# We have now substantially expanded the discussion to provide a detailed background of distributive justice and DEI and how they can be, or are, applied to health/medical research, and to incidental findings..

The distributive justice principles are rooted in the Rawisian theory. A statement has now been added to the description of the framework.

Page 6: Line 249

“Notably, the EUSTICE framework is rooted in the Rawlsian theory of distributive justice.”

3. Ethical aspects concerning incidental findings in neuroscience research

Researchers may have a responsibility to disclose any IFs to participants, to handle sensitive medical information with care, and to consider the potential broader impact of their research on the scientific community. By taking these considerations into account, researchers can ensure that their work is conducted ethically and responsibly. Overall, handling IFs in a fair and just manner involves treating all patients with respect and dignity and providing them with the necessary information and resources to make informed decisions about their healthcare. There are several key considerations surrounding principles of human research ethics that can guide researchers in handling IFs [26]: (a) Respect for autonomy, Informed Consent and Confidentiality: Researchers have a duty to respect the autonomy of research participants and to consider their interests and preferences when making decisions about how to handle IFs. Patients should be fully informed about the potential risks, benefits, and alternatives to any medical procedure or examination, including the possibility of IFs. This helps ensure that patients are able to make informed decisions about their care. The confidentiality of incidental findings should be respected, and the information should only be shared with those who need to know, (b) Nonmaleficence and Disclosure: Researchers have a duty to do no harm and should consider the potential risks and benefits of disclosing IFs to participants. This includes not withholding information that may be of benefit to research participants. It is important to disclose incidental findings to the patient in a timely and appropriate manner. This may involve discussing the findings with the patient, providing them with written information, providing appropriate follow-up care or referring them to a specialist for further evaluation, (c) Beneficence: Researchers have a duty to act in the best interests of research participants and should consider whether disclosing IFs would be beneficial to the participant, and (d) Justice and Quality of Care: Researchers should ensure that the handling of IFs is fair and equitable and that any decisions made are not biased or discriminatory. This includes ensuring that the benefits and burdens of research are distributed fairly among all participants. Patients should receive high-quality care regardless of their ability to pay or other social factors. Consider the potential financial impact of follow-up care and treatment on the patient, and work with the patient to identify options for financing this care.

Before introducing the EUSTICE framework, the following sections will provide a background on distributive justice and DEI concepts, and their application to human/medical research and ethics, which is at the foundation of the proposed framework.

Distributive justice: Distributive justice is a concept in political philosophy that refers to the fair and equitable distribution of resources, benefits, and burdens within a society [27]. It is a fundamental aspect of any just and fair society, as it determines how the benefits and burdens of social, economic, and political systems are distributed among the members of a community. There are various theories of distributive justice that attempt to address the question of how resources should be distributed within society. One of the most well-known theories is Rawls' theory of justice [28], which states that the distribution of resources should be such that it would be accepted by rational actors in an original position of equality [29]. Rawls argues that, in order to determine a fair distribution of resources, we must imagine ourselves in a hypothetical "original position" where we are free and equal, but do not know our own place in society, our natural abilities, or our social status. From this position, we would choose the principles of justice that would govern the distribution of resources in our society, including health and social determinants of health [30]. According to Rawls, the principles of justice that we would choose would be: (a) Each person is to have an equal right to the most extensive total system of equal basic liberties compatible with a similar system of liberty for all, and b) Social and economic inequalities are to be arranged so that they are both: (a) reasonably expected to be to everyone's advantage, and (b) attached to positions and offices open to all. Another theory of distributive justice is the libertarian theory [31], which states that individuals have a natural right to own and control their own property, and that the state has a limited role in redistributing wealth. According to this theory, the state should not interfere with individuals' rights to own and control their own property, and any redistribution of wealth should be voluntary, rather than mandatory. There are also various other theories of distributive justice, such as the socialist theory, which advocates for the collective ownership of the means of production and the distribution of resources according to the needs of the community; the Rawlsian difference principle, which argues that social and economic inequalities should be allowed as long as they benefit the least advantaged members of society; and the capabilities approach, which focuses on the abilities and opportunities that individuals have to lead a fulfilling life, rather than simply the distribution of resources.

The concept of distributive justice is closely related to the field of human research ethics, as it deals with the fair distribution of the benefits and burdens of research among different groups of people [32]. In order to ensure that human research is conducted ethically, it is important to consider the principles of distributive justice and ensure that the distribution of benefits and burdens is fair and just. One way in which the principles of distributive justice can be applied to human research ethics is by ensuring that the benefits of research are distributed fairly among different groups of people [33]. For example, if a particular research study is being conducted on a new medical treatment, it is important to ensure that the treatment is made available to all those who need it, regardless of their social or economic status. This means that the treatment should not be restricted to certain groups of people, such as those who can afford to pay for it, while others are left out.

Another way in which distributive justice can be applied to human research ethics is by ensuring that the burdens of research are distributed fairly among different groups of people. This includes ensuring that the risks of participating in a research study are minimized, and that participants are fully informed about the potential risks and benefits of participating in the study. It is also important to ensure that the selection of research subjects is fair, and that subjects are not unfairly disadvantaged or excluded based on their social or economic status. In addition to these considerations, it is important to ensure that the distribution of benefits and burdens from research is transparent and open to public scrutiny. This helps to ensure that the distribution of benefits and burdens is fair and just, and that any potential inequities are identified and addressed. Overall, the concept of distributive justice is an important consideration in the field of human research ethics, as it helps to ensure that the benefits and burdens of research are distributed fairly among different groups of people. By applying the principles of distributive justice to human research, we can help to ensure that research is conducted in an ethical and just manner, and that the benefits of research are shared fairly among all members of society.

Diversity, Equity & Inclusion:  Diversity, equity, and inclusion (DEI) theory is a framework that aims to promote fairness and justice in organizations and communities by promoting the inclusion and representation of diverse groups of people [34]. Diversity refers to the range of differences among individuals, including differences in race, ethnicity, gender, sexual orientation, age, ability, religion, and cultural background. Equity refers to the fair treatment of individuals and groups, including the fair distribution of resources, opportunities, and privileges. Inclusion refers to the active engagement and welcoming of diversity within an organization or community.

DEI theory recognizes that diversity and inclusion are essential to creating a just and equitable society, as they help to promote understanding, respect, and appreciation for differences among individuals and groups. DEI theory also acknowledges that historically marginalized and underrepresented groups often face barriers and discrimination in accessing resources, opportunities, and privileges, and therefore efforts to promote equity are necessary to address these inequalities [35]. It also involves acknowledging and celebrating the unique perspectives, experiences, and contributions of all individuals.

To promote DEI, organizations, and communities can adopt policies and practices that actively seek to increase diversity, promote equity, and create inclusive environments. This may include things like implementing hiring and promotion practices that promote diversity, providing resources and support for marginalized groups, and creating safe and welcoming spaces for all members of the community. By adopting DEI principles and practices, organizations and communities can work towards creating a more just and equitable society for all.

The principles of DEI can be applied to the field of human research ethics in order to ensure that research is conducted in a manner that is fair and just, and that the benefits of research are shared equitably among all members of society [36]. One way in which DEI principles can be applied to human research ethics is by ensuring that the selection of research subjects is diverse and representative of the population being studied [37]. This includes considering factors such as race, ethnicity, gender, sexual orientation, age, ability, and socio-economic status in order to ensure that the research sample is representative of the larger population. It is also important to ensure that underrepresented groups are not unfairly excluded from participating in research.

In addition to ensuring diversity in the selection of research subjects, it is also important to ensure that the benefits of research are shared equitably among all members of society. This may involve making the results of research available to all, regardless of their social or economic status, or ensuring that the benefits of research are distributed in a manner that is fair and just. DEI principles can also be applied to the design and conduct of research studies in order to ensure that the research process is inclusive and respectful of the needs and perspectives of all research subjects [37]. This may involve consulting with and involving underrepresented groups in the design and conduct of research, and ensuring that the research process is transparent and open to public scrutiny. Overall, the principles of DEI can be applied to the field of human research ethics in order to ensure that research is conducted in a fair and just manner and that the benefits of research are shared equitably among all members of society. By applying DEI principles to human research, we can help to create a more diverse, equitable, and inclusive society [38].”

Reviewer 3 Report

Through this manuscript, the author proposes a conceptual framework for handling incidental findings in brain imaging research, which takes into account  diversity, equity, inclusion and distributive justice approaches. This approach is clearly described and is an interesting proposal for researchers involved in neuroscience studies, with potential benefits for research participants, and particularly vulnerable groups.

The author could consider the following element in the proposed framework: In light of ongoing/future scientific research and novel findings, the clinical significance status of certain incidental findings may change. So the question that arises is whether the proposed IFs Committee will have a certain policy on this issue and whether there an ethical obligation for the IFs Committee to regularly update the clinical significance status of past research studies and inform research participants accordingly? 

The above presented point is only a suggestion to the author, and the manuscript can be published in its current form.

Author Response

C#1: Through this manuscript, the author proposes a conceptual framework for handling incidental findings in brain imaging research, which takes into account  diversity, equity, inclusion and distributive justice approaches. This approach is clearly described and is an interesting proposal for researchers involved in neuroscience studies, with potential benefits for research participants, and particularly vulnerable groups.

The author could consider the following element in the proposed framework: In light of ongoing/future scientific research and novel findings, the clinical significance status of certain incidental findings may change. So the question that arises is whether the proposed IFs Committee will have a certain policy on this issue and whether there an ethical obligation for the IFs Committee to regularly update the clinical significance status of past research studies and inform research participants accordingly? 

The above-presented point is only a suggestion to the author, and the manuscript can be published in its current form.

Reply# I thank the reviewer for a positive review of our work. As per the suggestion, we have added the following statement in the description of the framework (Page 8). Thank you for this valuable suggestion.

The adjudication or discussion on the clinical significance of IFs should be based on the standards of care or current recommendations or indications at the time of review of the merit of the case by the IF committee
